# Breast Tumor Segmentation in Dynamic Contrast-Enhanced MRI via Multi-Staged Training and Deep Ensembling of a Large Kernel MedNeXt

## 1 Introduction

Breast cancer is a major health concern [8], and Magnetic Resonance Imaging (MRI) plays an important role in its assessment, preoperative staging and treatment [14, 7]. T1-weighted dynamic contrast-enhanced (DCE) MRI can highlight tumor vascularity by using contrast agents, aiding in the localization of tumor lesions. Accurate segmentation of tumor boundaries on these images is clinically valuable, as it enables quantitative evaluation of tumor size, shape, and volume over time [16, 1]. High-quality tumor segmentation further facilitates advanced analyses such radiomics feature extraction [17] for other downstream tasks [2, 4], including pCR assessment [13] and the characterization of tumor types [5].

Manual biomedical image segmentation is highly time-consuming, and tends to suffer from inter- and intra-annotator variability sbject to the level of experience of a radiologist. Deep learning based automated segmentation methods have proven to be reliable in addressing the stated problems [9, 15, 12, 11], with some limitations in terms of data, and or architectural constraints. To achieve robust and generalizable performances, highly parameterized deep learning models need to be trained with sufficient enough labeled data. Readily available large-scale labeled medical imaging data is lacking, particularly for dynamic contrast-enhanced MRI segmentation.

We address the challenge of the segmentation of breast tumor lesions in multi-contrast MRI, using MedNeXt and a multi-staged training strategy of improving receptive field, and loss optimization based deep ensembling. Our segmentation method is motivated by the need to capture both the subtle enhancement patterns of tumors across multiple post-contrast time points and the broader breast tissue context, which larger receptive fields naturally accommodate. To effectively utilize large kernels without overfitting, we adopt a two-stage training strategy: we first train a MedNeXt model with the conventional $3 \times 3 \times 3$ kernel sizes, and then expand to $5 \times 5 \times 5$ kernel sizes via trilinear interpolation [15].

## 2 Methodology

### 2.1 Dataset and Preprocessing

We used a multicenter dataset of 1506 cases from over 20 institutions for training, and a held out 58 cases for testing [6]. Each case in the dataset includes a series of T1-weighted DCE-MRI volumes acquired at multiple time points: one pre-contrast and up to five post-contrast phases. For our experiments, we selected the pre-contrast image and the first two post-contrast images of each case.

The dataset comprises both unilateral and bilateral breast DCE-MRI scans. Data preprocessing, training and inference were done using the standard nnU-Net pipeline [9]. During training, we adopted nnU-Net's patch-based sampling strategy with a fixed input size of $128 \times 128 \times 128$ voxels.

Submitted to 39th Conference on Neural Information Processing Systems (NeurIPS 2025). Do not distribute.

To ensure sufficient exposure to tumor regions across both scan types, more than one-third of the patches in each batch were enforced to contain at least one foreground voxel. At inference time, segmentation was performed using overlapping sliding-window patches, ensuring full-volume coverage regardless of laterality.

## 2.2 MedNeXt Architecture

MedNeXt is a fully ConvNeXt [10] encoder-decoder U-shaped network for biomedical image segmentation [15]. Its inverted bottleneck design in the up and downsampling layers, and the compound scaling of depth, width and receptive field, makes it a highly capable segmentation method. It is further transformer-inspired in its scaling approach, and the use of large-kernel sizes to approximate attention. The added inductive bias provides the benefits of both convolutional- and transformer-based approaches, in capturing short and long range dependencies respectively.

$$M^{(5)} = UpKern(M^{(3)}, \text{size} = 5) \tag{1}$$

The approximation of attention via larger kernel sizes of $5 \times 5 \times 5$ instead of the conventional $3 \times 3 \times 3$ sizes is achieved by first pretraining a conventional MedNeXt ($M^3$), and trilinearly interpolating its convolutional kernels to an initialized large-kernel MedNeXt ($M^5$), using an algorithm called *UpKern* [15] in Equation 1. The performance saturation usually observed with increasingly large kernel sizes is mitigated [3].

## 2.3 Training Strategy

All networks were trained with deep supervision, stochastic gradient descent (SGD) optimization, and a cosine annealing learning rate schedule initialized at $1 \times 10^{-4}$, using an A100 NVIDIA GPU. For the base $M^3{}_{Base}$, training was conducted using a five-fold cross-validation split and optimized using Dice cross-entropy loss. Each fold was trained independently for 250 epochs. Following the completion of cross-validation, we identified the fold that achieved the highest mean Dice coefficient on its respective validation set, whose weights were used in the second stage of training.

In the subsequent stage, we employed the *UpKern* strategy 1 to resample the learned $M^3{}_{Base}$ convolutional kernel weights into $M^5{}_{Base}$ via trilinear interpolation. This approach enabled a smooth transition to a large-kernel configuration, thus expanding the effective receptive field of the network without introducing instability often associated with training large kernels from random initialization [3]. The newly initialized $M^5{}_{Base}$ was then fine-tuned for making use of the entire training set, with all other architectural and training settings held constant.

Additionally, we generated a second $M^5$ by applying the *UpKern* algorithm and fine-tuning the pretrained weights, forming $M^5{}_{Focal}$. In this stage, the network was optimized using a composite loss that combines Dice–cross-entropy and focal loss to better penalize small lesion segmentation errors and address class imbalance arising from large foreground–background differences:

$$\mathcal{L}_{\text{total}} = 0.25\,\mathcal{L}_{\text{Dice-CE}} + 0.75\,\mathcal{L}_{\text{Focal}} \tag{2}$$

where $\mathcal{L}_{\text{Dice-CE}}$ denotes the combined Dice and cross-entropy loss used in the first training stage, and $\mathcal{L}_{\text{Focal}}$ is the focal loss component that increases the weighting of hard-to-segment regions. Code is publicly available along with the implemented composite loss functions[1].

## 3 Results

Segmentation performance was evaluated on the held-out testing set. The results are summarized in Table 1. $M^3$ achieved a Dice score of 0.64 and a normalized Hausdorff Distance (NormHD) of 0.3. Upon applying the *UpKern* strategy and fine-tuning the best performing single model, $M^5{}_{Base}$ improved the Dice score to 0.66 and the NormHD to 0.29 (see Figure 1. Further ensembling with $M^5{}_{Focal}$ led to minimal increase in Dice to 0.67, and NormHD to 0.24. The reported baseline is a 5-fold nnU-Net ensemble by [6], also trained on the 1506 training dataset and evaluated on the test set.

---

[1] `https://github.com/toufiqmusah/caladan-mama-mia`

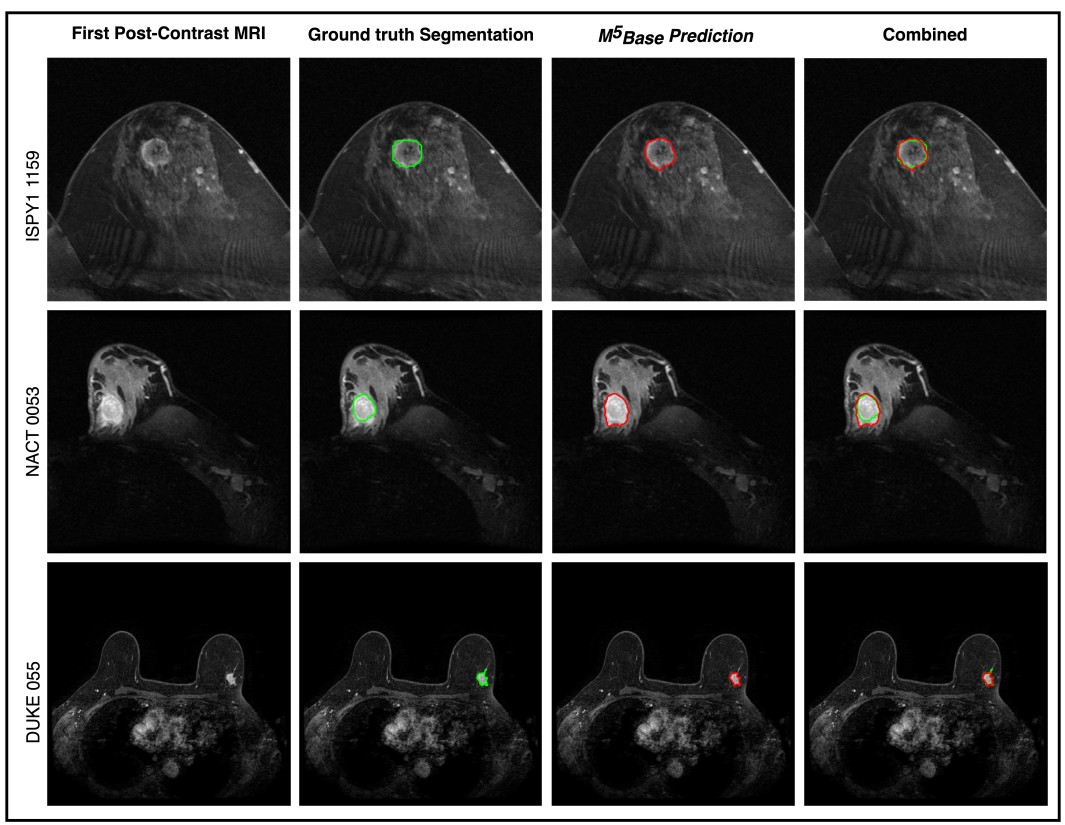

Figure 1: Qualitative segmentation performance of $M^5{}_{Base}$ on validation samples from various centers in in red, and ground truth in green.

Table 1: Performance comparison between methods. Metrics are reported as Dice Score and 95th-Percentile Normalized Hausdorff Distance. Total number of combined parameters are denoted in in millions.

| Architecture | Dice Score (↑) | NormHD (↓) | Parameters (↓) (M) |
|---|---|---|---|
| nnU-Net (5-Folds) [6] | 0.65 | 0.30 | 700 |
| $M^3$ (5-Folds) | 0.64 | 0.30 | 154 |
| $M^5{}_{Base}$ | 0.66 | 0.29 | **32.1** |
| $M^5{}_{Base} + M^5{}_{Focal}$ | **0.67** | **0.24** | 64.2 |

The 5-fold $M^3$ ensemble's performance improved after applying the *UpKern* strategy to expand the receptive field. Fine-tuning the single best-performing fold into $M^5{}_{Base}$ led to a Dice score of 0.66 and a reduction in NormHD to 0.29. Further ensembling $M^5{}_{Base}$ with $M^5{}_{Focal}$, which was trained using Dice-cross-entropy and focal loss, resulted in a Dice score of 0.67 and a NormHD of 0.24. These results validate the hypothesis that larger receptive fields improve segmentation performance by capturing broader anatomical context. Notably, both $M^5{}_{Base}$ and $M^5{}_{Base} + M^5{}_{Focal}$ outperformed the nnU-Net baseline [6] and achieved a lower NormHD, despite having substantially fewer parameters per model (32.1M vs. 140M per model instance) and using only two models in the final ensemble compared to the five-model nnU-Net baseline. Architectural efficiency and targeted loss function strategies can deliver improved performance while reducing computational requirements. Our study is limited in scope to the evaluation of the proposed *UpKern* strategy within the MedNeXt architecture; we did not test its generalizability across other network families. While ensemble performance was reported, we did not isolate and report the standalone performance of individual models within the ensemble, which could provide further insights into complementarity.

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
