# OpenReview forum: "Breast Tumor Segmentation in Dynamic Contrast-Enhanced Magnetic Resonance Images via Multi-Staged Training and Deep Ensembling of a Large Kernel MedNeXt"
_EurIPS.cc/2025/Workshop/MedEurIPS — EurIPS 2025 Workshop MedEurIPS Submission_

### Official Review · Reviewer_2tbh · 2025-10-27
**The author proposes a MedNeXt based-approach using trilinear interpolation to expland convolutional kernels and fine-tune it for the dataset. They claim it to performance better on Dice and NormHD over nnU-Net while being efficient in terms for parameters.**

**Rating:** 4
**Confidence:** 5

**Review:**

**Strengths**
- Important clinical problem and solution - an efficient automated segmentation of breast tumors
- Reports better performance and is also more parameter-efficient than the current SOTA for automated 3D medical segmentation.
- Code availability

**Weakness**
- The baseline (as cited) does not mention the performance referred to by the author. Moreover, the referenced nnU-Net model seems to be trained on a private dataset with a distinct setup (training split).
- No novelty regarding the kernel expansion, as already mentioned in the original paper.
- nnU-Net baseline is 5-fold mean and not ensembled. This makes the reported model size and comparison misleading.
- No validation rigor or statistical tests are reported. And, results are given as single-point Dice and NormHD values without variance/CI, making claims weak.
- Figure 1 is qualitative but lacks interpretation related to the study.

---

### Official Review · Reviewer_XX1s · 2025-10-29
**This paper applies an existing method to an existing dataset without clear contribution.**

**Rating:** 3
**Confidence:** 5

**Review:**

This paper applies MedNext network to a public breast cancer segmentation dataset.

The main issue with this work is the absence of clear contribution, as it seems the authors
only applied an existing method to an existing dataset, without much originality.

The quality of the manuscript is not satisfying: no abstract (?), no figure to visualize the method,
confusion between result, discussion. No conclusion. Moreover, I have a doubt
about the reported 0.65 DSC metrics for the baseline method, as in the original paper
0.65 is the IoU and not the DSC which is around 0.75.

Pros: large dataset (but public); similar results than nnUNet but at a lower computational cost (but
the method, MedNext is not theirs).

Cons: Contribution? Abstract? Discussion? Conclusion? Likely confusion on the validation metric.
Ill posed problem in the introduction as the second paragraph ends with the idea of data scarcity,
but then the rest of the paper has nothing to do with.

---

### Decision · Program_Chairs · 2025-11-03

**Decision:**

Reject

**Comment:**

Both reviewers find the topic clinically relevant but identify major weaknesses: the paper lacks clear novelty, methodological rigor, and proper validation.